# Smartphone Sensors for Monitoring Cancer-Related Quality of Life: App Design, EORTC QLQ-C30 Mapping and Feasibility Study in Healthy Subjects

**DOI:** 10.3390/ijerph16030461

**Published:** 2019-02-05

**Authors:** Sabina Asensio-Cuesta, Ángel Sánchez-García, J. Alberto Conejero, Carlos Saez, Alejandro Rivero-Rodriguez, Juan M. García-Gómez

**Affiliations:** 1Instituto de Tecnologías de la Información y Comunicaciones (ITACA), Universitat Politècnica de València, Camino de Vera s/n, 46022 Valencia, Spain; ansan12a@inf.upv.es (Á.S.-G.); carsaesi@ibime.upv.es (C.S.); juanmig@ibime.upv.es (J.M.G.-G.); 2Instituto Universitario de Matemática Pura y Aplicada, Universitat Politècnica de València, Camino de Vera s/n, 46022 Valencia, Spain; aconejero@upv.es; 3Salumedia Tecnologías, 41011 Sevilla, Spain; alejandrorivero@salumedia.com

**Keywords:** oncology, cancer, activity recognition, mobile sensors, pattern recognition, visual analytics

## Abstract

Quality of life (QoL) indicators are now being adopted as clinical outcomes in clinical trials on cancer treatments. Technology-free daily monitoring of patients is complicated, time-consuming and expensive due to the need for vast amounts of resources and personnel. The alternative method of using the patients’ own phones could reduce the burden of continuous monitoring of cancer patients in clinical trials. This paper proposes monitoring the patients’ QoL by gathering data from their own phones. We considered that the continuous multiparametric acquisition of movement, location, phone calls, conversations and data use could be employed to simultaneously monitor their physical, psychological, social and environmental aspects. An open access phone app was developed (Human Dynamics Reporting Service (HDRS)) to implement this approach. We here propose a novel mapping between the standardized QoL items for these patients, the European Organization for the Research and Treatment of Cancer Quality of Life Questionnaire (EORTC QLQ-C30) and define HDRS monitoring indicators. A pilot study with university volunteers verified the plausibility of detecting human activity indicators directly related to QoL.

## 1. Introduction

Smartphones currently represent a significant and growing presence in people’s daily lives worldwide. The number of phone users in the world is forecast to grow from 2.1 billion in 2016 to around 2.5 billion in 2019. Android, with 80% of all phones sales, leads the market. In contrast, about 15% of all the phones sold to end customers have iOS as their operating system [1]. The widespread use of phones offers new opportunities for the development of mobile apps in areas such as cancer healthcare [2].

Many phones are now equipped with various sensors, including accelerometers, Wi-Fi, light and temperature sensors, gyroscopes, barometers, etc. These sensors have become rich data sources for measuring various aspects of the user’s daily life. The typical physical activities include walking, jogging, sitting, etc. Due to their unobtrusiveness, low or no installation cost, and being easy to use, phones are becoming the leading platform for human activity recognition [3].

The most popular sensors are those related to user location and allow the provision of mobile services that are tailored to the user’s location, namely location-based services [4]. Other sensors, like accelerometers, have been used in activity recognition tools, within the popular area of activity trackers and fitness apps [5].

The above examples are related to general-purpose applications that facilitate and improve the user’s life in a certain way, but they are not medical applications, nor are they widely used by healthcare professionals. According to Higgins [6], healthcare providers can benefit from the development of those mobile services based on sensor data; phone apps can be used to monitor users’ health or fitness parameters, by using built-in sensors such as the accelerometer, microphone, speaker, and camera. For instance, this type of application can be used by healthcare professionals to monitor patients between visits [6]. Further, researchers are now tapping into using smartphones for medical diagnostics such as [7,8].

The present trend is towards applications which can be used to collect data on patients’ quality of life (QoL) [9]. Studies have been published on the feasibility of using sensor data to monitor mobile phone users’ QoL. A recent study showed the feasibility of using passively collected mobile phone keyboard metadata to predict signs of manic depression via clinician-administered rating scales [10]. This could be promising for cancer patients, due to the prevalence of depression in this group [11].

In this context we developed the Human Dynamics Reporting Service (HDRS) and here propose it as a mHealth solution to assist cancer patients. The system is based on the multiparametric continuous acquisition of movement, location, phone calls, conversations and use of data to simultaneously monitor the physical, psychological, social and environmental activity of patients undergoing oncological treatment. These features are incorporated in the phone application known as HDRS. HDRS monitoring indicators were also defined and mapped with standardized QoL items for cancer patients, the European Organization for the Research and Treatment of Cancer Quality of Life Questionnaire (EORTC QLQ-C30). A pilot study was conducted on university volunteers to validate the detection of physical and social activity indicators directly related to QoL.

### Other New Approaches in Monitoring Cancer Patients’ Quality of Life

The World Health Organization defines QoL as a multi-dimensional concept formed by the physical, psychological, social and environmental domains [12]. Chronic diseases such as cancer have a strong influence on both physical health and QoL, which together comprise the actual concept of health-related quality of life (HRQoL), the complete state of physical, social, and psychological functioning [13]. The evidence that QoL is a prognostic variable for cancer patient survival has led to its increased routine incorporation into clinical research [14], so that improving survival rates and preserving the patient’s HRQoL [15] has become a major goal in the treatment of cancer.

Patient-reported outcomes (PROs) play an increasingly important role in complementing clinical outcome parameters in measuring HRQoL. In fact, PROs are currently the accepted gold standard for collecting data about patients’ subjective perception of their own state of health [16]. One of the most significant PROs for HRQoL measurement in cancer patients is provided by the European Organization for the Research and Treatment of Cancer (EORTC) [17], the quality of life questionnaire QLQ-C30 version 3.0 (EORTC, Brussels, Belgium) [18].

As a result of technological advances, apart from the phone sensing applications already mentioned, other approaches have been proposed to assist patient’s health, such as electronic patient-reported outcomes (e–PRO), wearable fitness trackers and action-monitoring or actigraphy. These approaches facilitate the daily monitoring of patients’ health and could reduce the current barrier to continuous monitoring of cancer patients in clinical trials. The new actigraphy devices in particular provide non-intrusive and non-stigmatizing monitoring of outpatients for weeks or even months and facilitate treatment outcome measure during daily activities [19].

Applications such as e–PRO installed on the patients’ own phones can be employed to collect QoL data remotely [9]. Systems like e–PRO are used in oncology clinical care because of their ability to: (i) enhance clinical care by flagging essential symptoms and saving clinician’s time; (ii) improve the availability of standardized methods for creating and implementing PROs in clinics; and (iii) facilitate the access of user-friendly platforms for patient self-reporting, such as tablets, computers and automated telephone surveys [20]. Recently, Wallwiener et al. (2017) [21] evaluated an e–PRO version of the PRO EORTC QLQ-C30 and concluded that it was reliable for patients with breast cancer, showing a high correlation in almost all questions (and on many scales).

New developments in wearable activity monitors mean they can now collect objective patient activity data in a non-obtrusive manner [22]. Continuous physical and physiological monitoring in any environment would shorten hospital stays, improve both the recovery and reliability of diagnosis and improve patients’ QoL. Apart from physiological measurements, the daily physical activity of chronic patients reflects the quality of their daily lives [23].

## 2. Materials and Methods

In this section we describe both the HDRS application and the pilot study designed to test its feasibility as a potential tool for monitoring cancer patients, through movements, location, conversations, and data using sensors in their phones. First, we give the technical data of HDRS and explain the workflow of the web application in charge of registering users and saving the data obtained. We then describe the participants and data collection (definitions of subjects’ characteristics and prediction variables). A hypothesized association between HDRS monitoring variables and the constructed scales of the EORTC QLQ-C30 Version 3.0 [18] is also given.

### 2.1. Mobile Application Human Dynamics Reporting Service (HDRS)

This section gathers information on the application requirements functionality and workflow and explains the workflow of the web application in charge of registering users and saving the data obtained from them.

#### 2.1.1. Requirements

In order to obtain the necessary physical and social information for the functioning of the application, it is necessary to have access to the following sensors:Accelerometer: obtains the data related to the user’s movements.Microphone: obtains the data related to the sound surrounding the user.In order to run the application correctly, the phone must have an Android operative system with the sensors mentioned above, plus Wi-Fi + 4G to obtain the data related to location, and also a sim card for calls.

#### 2.1.2. Application Workflow

We here describe the different steps in the HDRS workflow:Agreement: A screen with the study conditions is displayed to the user before starting. This is the electronic adaptation of the "participant informed consent model" provided by our university, Universitat Politècnica de València (UPV). If the user does not accept the conditions by clicking on the accept button, he will not be able to go to the Boot screen.Boot: The app can be booted in two ways:
○When it is initialized and the user is not registered, the registration form is shown.○When the app is booted and the user is already registered, the agreement menu is shown.Registration Form: asks the user for the following information (Figure 1):○Gender: Male or female.○Weight (kg): The weight of the user in kg.○Height (cm): The height of the user in cm.○Country: Country of residence.○Occupation: The daily physical activity of the user depending on his occupation.○Level of activity/exercise: The workout activity related to the user’s activities.Permissions: The following permissions are required for running the application properly:○Enabling the app to run in the background.○Enabling microphone use.○Access to call register.○Access phone location.○Acceptance of Agreement shown when the application is booted.Utilities: We included the following functions to facilitate data acquisition and processing:○Background: When the app is running in the background, a notification is shown for the user to stop the application, synchronize data to the server, or tag an event.○Tagging an event: When the user is accessing the tag event action from the main menu, a message dialog is displayed to add the description of the current event (Figure 2).○Synchronize data: When the user is accessing data sync from the main menu, a message is shown when data is sent and another to confirm end of sync.○Closing the app: When closing the app from the menu, the service will finish and the application will close.

#### 2.1.3. Web Application

The web application functions are: (1) registering users on the database; (2) storing user data obtained from the application; and (3) showing the users’ table.

The web service is built in Ruby, using the Sinatra framework to build the web application to operate with the database and users’ data.

The registration process is made on the application, and the values are sent to be stored in a database on the web. The information obtained by the sensors is also sent to the web application. When it is received, this information is stored in the Google Drive folder created for this app use.

### 2.2. Data Colletion

Ethical approval was obtained for the study from the Ethical Committee of the Universitat Politècnica de València (UPV, Ethical Code: P8_12_11_2018). The subjects were volunteers from the Instituto de Tecnologías de la Información y Comunicaciones (ITACA) recruited face-to-face and by email invitations. All the participants gave their consent via the app.

#### 2.2.1. Participants

The participants received a brief explanation of the study objectives and the particular purpose of the HDRS application. They were also taught how to use the app functions; agreement, registration data, event tagging, and data synchronization. A total of five volunteers participated in the study. 

The average age of the participants was 36.2, with a maximum of 45 and minimum of 21.

#### 2.2.2. Sensor Data Collection

The following sensors were used to collect data for our purpose:The accelerometer obtained the linear acceleration of the device with a vector of three components (using the XYZ axis) excluding the value of gravity to give more realistic results related to movements. The values of each axis were measured in m/s^2^. When the data was saved to a CSV file, it computed the amount of movement by calculating the magnitude of the vector. The final result is measured in m/s^2^.Internet usage is the amount of data received and sent through the device measured in Megabyte. The time started when the application was booted.The microphone captured the frequency of the sound and calculated the average frequency of 100 sound samples obtained in 5 s to detect conversation. If the average sound frequency in one minute was greater than 50 Hz, then the value given was 1. If there was no noise it was saved as a low frequency near 0 or −1.Position calculated the location of the user using Wi-Fi and cell tower networks. The values may have had an offset of 25 m and were saved as latitude and longitude.Calls sensor registered calls in and out.

The sampling frequencies considered for the sensors were the following:Accelerometer: The number of movements was calculated every 60 s.Internet usage: The amount of data sent and received was calculated every 60 s.Microphone: To detect conversations it recorded 5 s of audio, obtaining 100 samples of sound frequency. This measurement was repeated every 60 s.Location: Since the position depended on the API used, the minimum sampling time was 60 s, but may have been longer.Phone calls: Calls in and out were calculated every 60 s.

#### 2.2.3. Defined HDRS App Monitoring Indicators

A total of six monitoring indicators were defined and calculated from the data collection sensors:Amount of movement (m/s^2^), provided by the accelerometer sensor. Movements per hour were calculated by summing all the samples in a minute by the user’s accelerometer sensor.Distance from a specific place (km) by the location sensor. The distance travelled represented the number of meters travelled per hour by the user by summing all the samples in a minute.Talk indicator was provided by the microphone. We assigned the value "1" if a conversation or other sources of sound were obtained and “0” if not, in minutes per hour.Data traffic (sent data + received) Mb provided by the smartphone. The mobile data traffic indicator represented the number of megabytes consumed per hour and was calculated as the sum of the received and sent data in Mb every minute.Number of phone calls (number of calls/min) made + received in a minute.User activity tag events.

#### 2.2.4. Data acquisition protocol

Sensor data from the participants concerning the accelerometer, internet usage, microphone, location and phone calls was acquired for a period of seven days (24 h) with the acquisition frequency defined in Section 2.2.2 (mostly every 60 s). The participants sent the data obtained by the sensors to the web application once a day by the app´s synchronize data function (Section 2.1.2). On reception, this information was stored in the Google Drive folder created for the use of the app. The data could be accessed and downloaded for calculating defined HDRS app monitoring indicators with the user list URL (Section 2.2.3).

The participants who recorded less than three days of 24 h data acquisition from all the sensors were discarded from the pilot study.

## 3. Results

This section gives the results obtained from the pilot study with the HDRS application. First, we show each monitoring indicator from the data acquired by the HDRS app from the different sensors and then indicate the related EORTC QLQ-C30 questionnaire items. We also point out that other information on the user’s physical, cognitive, and social aspects can be obtained from them.

We also give a summary of the proposed mapping between HDRS monitoring indicators and items in the EORTC QLQ-C30, followed by the results of HDRS app usability test with the participants.

### 3.1. User Movements

Figure 3 shows a graph of the user indicator values monitored in a week. The increased number of driving, walking and working-out movements and the decreased activities such as working, eating, or watching TV can be seen thanks to the event tags entered. Figure 4 shows differences between participants in the movement indicator in one day. One of the participants was excluded from this figure due to a problem with the accelerometer and missing data. It should be noted that in general, lower movement values were recorded at night (22 pm to 7 am), while they increased during the rest of the day.

The user movements computed weekly can be applied to assess the following EORTC QLQ-C30 physical and role functioning items:Strenuous physical activities (Item 1 of EORTC QLQ-C30). This item is associated with a high number of movement values. In Figure 3 working out is indicated and the corresponding increase in movements.Long/short walk (Items 2 and 3 of EORTC QLQ-C30). Here, the amount of movement would be higher than in sedentary activities such as watching TV.Need to stay in bed (Item 4 of EORTC QLQ-C30) or sitting at home. In this case, the number of movements will be very few, close to nil.Limited in work or other daily activities; limited in hobbies/leisure time activities (Items 6 and 7 in EORTC QLQ-C30). Fewer movements are assumed than in regular activities.

### 3.2. Distance Traveled

Figure 5 shows all the distances travelled by a user in a week. The hours when the device is kept still are not shown on the chart. Longer distances were registered when driving or walking. Figure 6 shows the differences between the participants in distance travelled in one day. Late at night no distances were registered. In general, long distances in one hour were due to vehicle transfers.

Apart from the registered tagged events, the user’s speed could be used to determine if the subject was walking or running. For example, the mean comfortable gait speed ranged from 127.2 cm/s for women in their seventies to 146.2 cm/s for men in their forties. The mean maximum gait speed ranged from 174.9 cm/s for women in their seventies to 253.3 cm/s for men in their twenties [24].

The distance-travelled weekly indicator could also be associated with the following EORTC QLQ-C30 physical and role functioning items:Strenuous physical activities (Item 1 of EORTC QLQ-C30). A long-distance walk or run could be considered as strenuous physical activity.Long/short walk (Item 2 and 3 of EORTC QLQ-C30). Tags added by the user were useful to identify when the user was walking, while the distance-travelled indicator could distinguish short from long walks.Need to stay in bed (Items 4 of EORTC QLQ-C30). This indicator, combined with data from other sources, may be associated with no distance travelled.Limited work or other daily activities; limited hobbies/leisure time activities (Items 6 and 7 in EORTC QLQ-C30) were associated with a significant decrease in the distance travelled compared with other reference periods of regular activity.

### 3.3. Mobile Data Traffic and Conversation

Figure 7 shows the data traffic (Mb) and conversations of a user in a week. Conversation is frequent when eating, working, walking, and watching TV, showing the relationship between conversation and social activities, although filters are needed to differentiate between conversation and sounds from the TV or the environment. Figure 8 shows the differences between the participants’ data traffic in a day. In general, more data traffic is registered in the early afternoon (2–3 PM) and night (8 PM–11 PM). These periods could be associated with rest activities between 4 am and 8 am (central sleeping hours), when no data traffic was recorded.

The mobile data traffic indicator could help to identify the user’s activity if compared with standard phone consumption when viewing a video, listening to music, writing or talking in a forum, social networking, etc. This should also include data downloaded from a Wi-Fi connection.

The mobile data traffic and talk indicators (weekly) could be associated with social and cognitive functioning EORTC QLQ-C30 items:Cognitive functioning (Item 20 of EORTC QLQ-C30 scales). This was considered for determining focus/concentration difficulties. High hourly consumption of Mb may indicate that the user was concentrating on an activity such as writing, using apps, in social media, searching, reading, chatting and sharing photos, video, watching, etc.Social functioning (Items 26 and 27 of EORTC QLQ-C30). This could be used to find out if the physical condition and its relation to the medical treatment have interfered with the user’s family life and social activities. Regular social functioning should be associated with frequent conversations and data traffic, mainly from checking and participating in social networks.Limited work or other daily activities; limited hobbies/leisure time activities (Items 6 and 7 of EORTC QLQ-C30). These items were associated with reduced data traffic and talk frequency compared with the user´s periods of reference with a regular activity.

### 3.4. Number of Phone Calls

The number of calls in and out was considered a poor indicator. Phone calls have now been replaced by internet calls or chat messages. In the pilot study the mean number of calls was 3.4. While two users registered zero calls in a week, others registered four, six and seven.

### 3.5. Tag Events

A difference was detected between the event tags and the expected records of the sensors. The users indicated that they had forgotten to tag but did so once the action had started or even when it had finished.

### 3.6. Summary of Mappings between HDRS Monitoring Indicators and Items Of The EORTC QLQ-C30

The EORTC QLQ-C30 (Version 3.0) [18] questionnaire is composed of both multi-item scales and single-item measures. These include five functional scales, three symptom scales, a global health status/QoL scale, and six single items. Each of the multi-item scales includes a different set of items, with no item appearing in more than one scale. All the scales and single-item scores range from 0 to 100. A high scale score represents a higher response level. Thus, a high score for a functional scale represents a high/healthy level of functioning, a high score for the global health status/QoL represents a high QoL, but a high score for a symptom scale/item represents a high level of symptomatology/problems. The principle for scoring these scales is normalized as follows; firstly, by estimating the average of the items that contribute to the scale (this is the raw score); and secondly, by using a linear transformation to standardize the raw score, so that scores range from 0 to 100. A higher score represents a higher (“better”) level of functioning, or a higher (“worse”) level of symptoms [25].

The associations between HDRS monitoring indicators and items of the EORTC QLQ-C30 (Version 3.0) questionnaire are summarized in Table 1.

### 3.7. Calculating QoL Global Score from HDFS Monitoring Indicators

To obtain a global QoL score from the defined HDRS monitoring indicators, the following procedure was proposed according to the EORTC QLQ-C30 questionnaire [26]:

First, convert monitoring indicators associated with QoL items in a range between one and four to indicate frequency of occurrence. Such items are scored in EORTC QLQ-C30 questionnaire as; (1) “not at all”, (2) “a little” (3) “quite a bit” (4) “very much”. This was necessary to be able to validate the app comparing QoL scores obtained from it with the EORTC QLQ-C30 questionnaire scores. In the EORTC QLQ-C30 questionnaire each item has four response alternatives, except for the global health-status/quality of life scale, which has response options ranging from (1) “very poor” to (7) “excellent”. Then calculate the Global QoL Score according to Equation (1):
Global QoL Score = (Functional Scales + Symptom Scales + Global Health Status)/NS,(1)
with NS, the total number of scales;
Functional scales = PFS + RFS + EFS + CFS + SFS(2)
Symptom scales = FAS + NAS + PAS + DYS + IMS + APS + COS + DIS + FIS(3)
where PFS is the physical functioning score (movement, distance indicators). RFS is the role functioning score (movement, distance indicators). EFS is the emotional functioning score. CFS is the cognitive functioning score (data traffic indicator). SFS is the social functioning score (talk, data traffic, call indicators). FAS is the fatigue score. NAS is the nausea and vomiting score. PAS is the pain score. DYS is the dyspnea score. IMS is the insomnia score (talk, data traffic, call indicators). APS is the appetite loss score. COS is the constipation score. DIS is the diarrhea score. FIS is the financial difficulties score.

The principle for scoring functional and symptom scales are:
Estimate the average of the items (app indicators) that contribute to the scale; this is the raw score.
Raw Score (RS) = (I1 + … + In)/n(4)Use a linear transformation to standardize the raw score, so that scores range from 0 to 100; a higher score represents a higher (“better”) level of functioning, or a higher (“worse”) level of symptoms.
Functional Scales: Score = 1 − [(RS − 1)/range] × 100(5)
Symptom scales/items and Global health status/QoL: Score = [(RS − 1)/range] × 100(6)

Thus, a high score for a functional scale represents a high/healthy level of functioning, a high score for the global health status/QoL represents a high QoL, but a high score for a symptom scale/item represents a high level of symptomatology/problems.

It should be mentioned that future versions of the app will include symptoms and emotional function tagging and their corresponding app indicators. Global health status questions will also be included to calculate the global health status.

### 3.8. Usability Assessment (System Usability Scale Questionnaire)

To assess the usability of the HDRS app, the participants in the pilot study responded to the system usability scale questionnaire (SUS). SUS calculated a single score representing a measure of the overall usability of the system. To calculate the SUS score, we first added the score contributions from each item. Each item’s score contribution ranged from zero to four. For items 1, 3, 5, 7, and 9 the score contribution was the scale position minus one. For items 2, 4, 6, 8 and 10, the contribution is five minus the scale position. Finally, we multiplied the sum of the scores by 2.5 to normalize the final score between 0 and 100 [26].

Figure 9 shows USU scores by participant; only one participant has a score of 65, under the threshold of 68 points (50th percentile) [27], which represents an acceptable usability level. The USU mean score is 75, considered as good. However, in these preliminary results we are aware that the low sample size is a limitation of this usability study. Further usability improvements and assessments are planned.

## 4. Discussion

In a one week period, the HDRS application was shown by some users to be a feasible tool for monitoring user activities associated with QoL via their smartphone sensors. However, they reported problems regarding battery and memory consumption, forgetting tagging events and data synchronization. Two technical incidents were also registered; missing user registered data and pauses in the sensor data registry due to low batteries.

In this work preliminary relationships were set up between items associated with cancer patients’ QoL and the variables monitored by HDRS, although it is necessary to extend the study to validate these hypotheses. Future studies would include the evaluation of the QoL of patients undergoing oncological treatment by the EORTC QLQ-C30 questionnaire, HDRS monitoring for a week, as well as a subsequent analysis of the correlations between QoL items and HDRS indicators.

Apart from the EORTC QLQ-C30 questionnaire, other indicators related to QoL could be considered and linked with the HDRS app’s monitoring data and indicators, such as patterns of daily/weekly physical activity, regularity of physical activity, types of physical activity (walking, cycling, running), use of means of transportation, patterns of daily/weekly social activity, sleep periods, etc. In this way, nonlinear models can be considered for describing these patterns and for contributing to the improvement of cancer treatments [28].

In relation to the variables associated with user movement and distance travelled, it is important to highlight their relationship with the user´s physical activity. They are important because physical activity is perceived to be an effective non-pharmacological therapy in cancer patients by relieving the distress caused by physical and psychological symptoms. A growing body of evidence supports the idea that increasing physical activity provides important benefits by promoting psychological and physical well-being in cancer patients. Engaging in physical activity is related to improved HRQoL among breast cancer survivors [29]. Wearable fitness trackers, such as Fitbit, which measure daily steps may be useful tools to evaluate and help cancer patients’ treatments [30]. Bade et al. (2018) [31] recently showed the feasibility of using a Fitbit accelerometer to measure step counts in advanced-stage lung cancer patients. Correlations with high QoL, low depression, and low symptom scores with the more active patients were found. Such results can be relevant for both the physical and psychological domains of the QoL concept.

It is also possible to estimate user sleeping periods from HDRS data since the registers of HDRS indicators diminish to almost null during these periods. It is well known that sleep problems are a common cause of reduced QoL [32]. Different clinical issues can cause sleep problems, and often physical activity can be used to alleviate the symptoms [33]. Another critical factor, which is also linked to physical activity and sleep quality, is cancer-related fatigue, unless it is caused by mental health issues such as anxiety and distress. To monitor these aspects, it is common to use a set of methods collectively referred to as ecological momentary assessments (EMA) [34]. These are often complemented by wearable devices for tracking physical activity and sleep objectively [35].

On the other hand, while the promise of mHealth is that we can leverage the power and ubiquity of mobile and cloud technologies to monitor and manage side effects and treatment outside the clinical setting, it is essential to be attentive to usability and to keep the intended users, tasks and considered environments in mind [36]. The results obtained show the HDRS application utility as a potential tool for monitoring user activity. Although the tests carried out in the study are satisfactory from the perspective of the tool´s functional capacity, the future use of the app by cancer treatment patients requires an in-depth study of fundamental aspects such as user-centered design and usability. Adapting the app’s interaction to the specific characteristics and needs of cancer patients is one of the main prerequisites for developing a useful and effective mHealth application. At present, there are several proposals to help developers and researchers in the development, integration, and evaluation of mobile health services that support cancer patients in managing their health-related issues [2,37].

In addition to usability, many factors can influence how data is understood and can be used for making decisions. In the case of mobile and wearable health data, we face the additional challenges that in most cases the data can be sparse, hard to analyze and most healthcare professionals are not familiar with its use. Data-driven cancer care might require the development of tools looking into the whole concept of exposome informatics [38]. Another related challenge for the accomplishment of this vision is the semantic interoperability between clinical data and PROs/lifestyle data that can be tackled by the use of semantic health technologies to better capture the context of the patient using mobile and wearable technologies. In fact, the creation of ontologies to support knowledge-inferring from mobile and wearable devices has already been considered [39].

## 5. Conclusions

In this study we assessed the feasibility of monitoring the quality of life of cancer patients by phone sensors in a direct approach due to the need to monitor QoL as a clinical outcome in clinical trials for new cancer treatments. Most of the SoA (Service-Oriented Architectures) is now done using expensive/specific devices. Although the EORTC QLQ-C30 questionnaire has been previously adapted to the smartphone app format [40,41], and applied to assess the effects of a smartphone app intervention in QoL of patients with cancer [42], none of the previous research involved using the phone´s sensors to estimate QoL as defined in the EORTC QLQ-C30 questionnaire. Our approach was to map phone sensors to the EORTC QLQ-C3, which would bring a massive amount of technology within the reach of clinical trials, giving access to clinical outcomes of a large number of patients. Our study; (1) shows the feasibility of capturing relevant QoL information, (2) offers research groups an open access phone application for human dynamics reporting, (3) proposes the novel mapping between standardized QoL items for cancer patients (EORTC QLQ-C30) and HDRS monitoring indicators. The HDRS app is able to monitor users’ daily/weekly activities through indicators of their movements, distance travelled, conversations and data traffic. It also allows users to report their own activity by using the event-tagging app functionality. In addition, a graphic representation was developed for the indicators to facilitate their analysis.

The next step in this research program will define the specific method of converting phone sensor data to individual QoL item values defined in the EORTC QLQ-C30 and mapped in the present work. It will involve the acquirement of at least 200 samples following our smartphone sensor approach for evaluating QoL and the estimation of QoL items by multiple linear/non-linear regression of smartphone sensor data. Alternatively, we will calculate QoL items following a rule-based approach, e.g., Item 2 EORTC QLQ-C30: “Do you have any trouble taking a long walk?” could be calculated by following the steps: (1) Calculate the distance travelled in meters daily (dd) from the latitude and longitude records collected every 60 s by the location sensor. The Harvesine equation [43] would be used to calculate the value of the distance travelled. (2) Count the number of times the distance travelled daily is greater than or equal to the reference value of ‘long walk’ in meters according to EORTC QLQ-C30 in a week. If dd = 0, record value item 4 (“very much”); if dd = 1 item record value 3 (“remove a bit”); dd = 2 record value item 2 ("a little"); dd ≥ 3 record value item 1 (“not at all”). These values will be adjusted after experimentation and clinical judgment. A pilot study involving patients undergoing oncology will also be required to test the feasibility of HDRS monitoring and validate the proposed QoL indicators.

Moreover future work will also address the improvement of app design by including automatic recognition of user activities (walking, running, consulting, chatting, etc.), improving the app’s performance and the graphical interface under usability criteria, defining a set of activity tags to be selected by the user instead of writing text and a tag reminder pop-up, periodic auto-sending of recorded data, definition of new mobile sensor indicators associated with cancer patients’ QoL, the integration of graphs of QoL indicators into the app, and their adaptation to iOS. 

Finally, we are aware that the study is limited as regards the size of the participant’s sample and the length of the data acquisition period. Although the results are promising, larger samples will be needed to validate the technical issues and validate the defined indicators.

## Figures and Tables

**Figure 1 ijerph-16-00461-f001:**
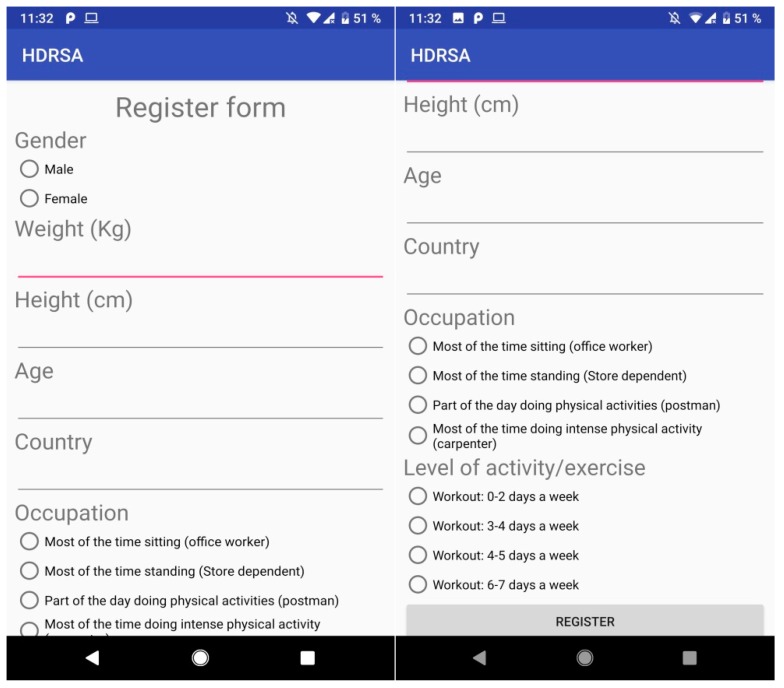
Mobile application Human Dynamics Reporting Service (HDRS) registration forms.

**Figure 2 ijerph-16-00461-f002:**
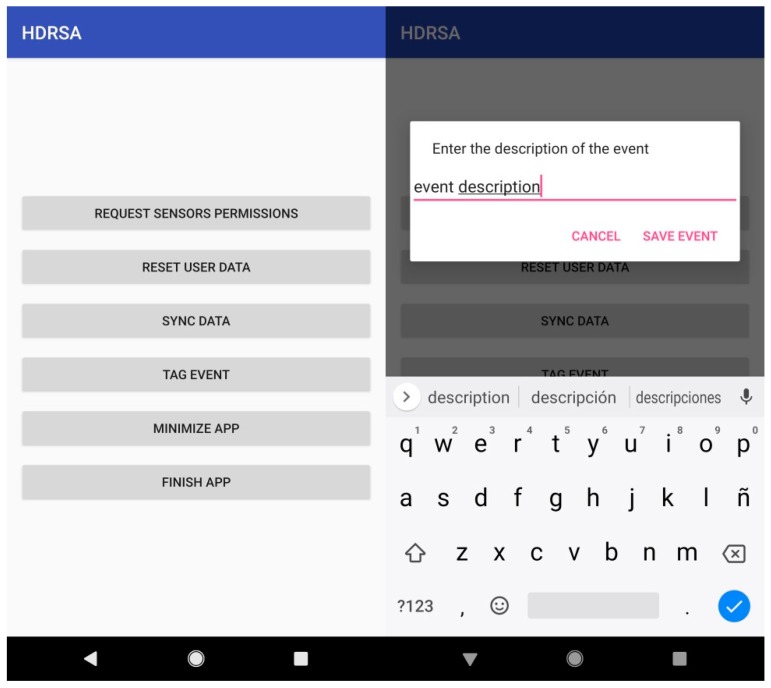
Mobile application HDRS tag event.

**Figure 3 ijerph-16-00461-f003:**
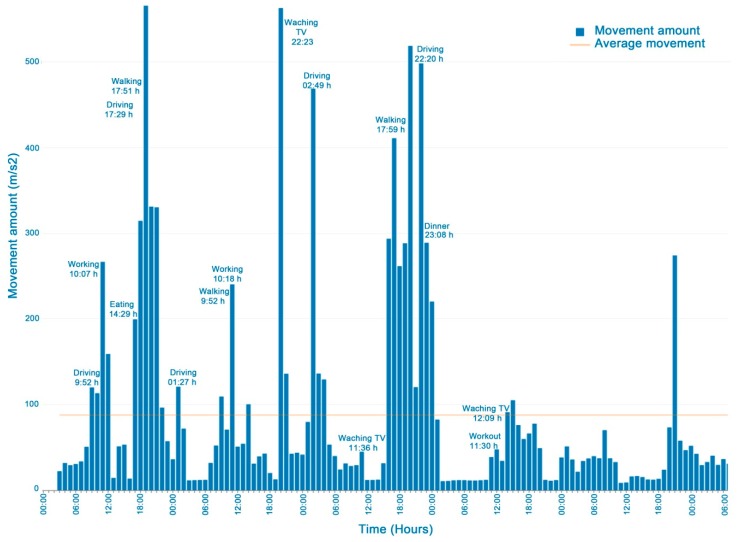
Amount of movement per hour and event tags in a five day HDRS monitoring period.

**Figure 4 ijerph-16-00461-f004:**
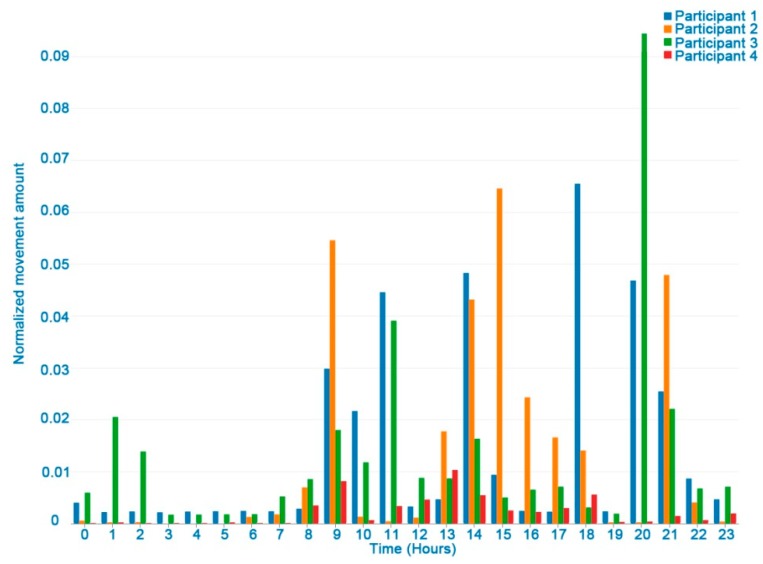
Normalized number of movements of participants in a one day HDRS monitoring period.

**Figure 5 ijerph-16-00461-f005:**
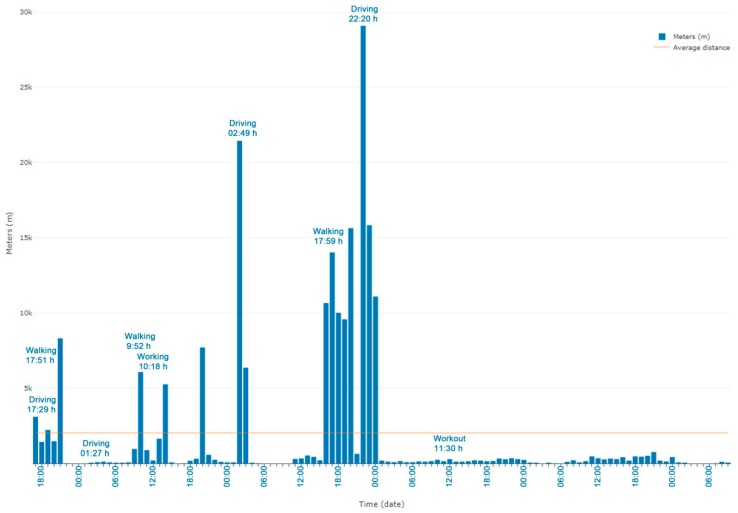
Distance travelled by the user (m/h) and event tags in a week of HDRS monitoring.

**Figure 6 ijerph-16-00461-f006:**
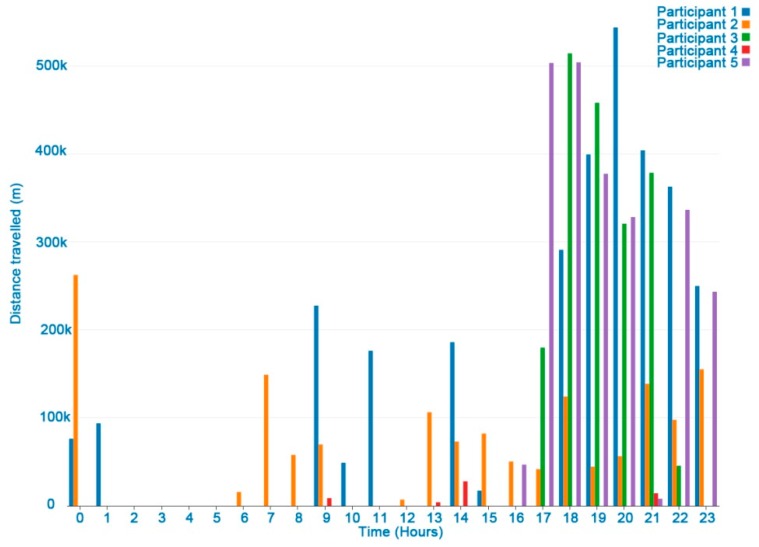
Distance travelled by participants (m) in a day of HDRS monitoring.

**Figure 7 ijerph-16-00461-f007:**
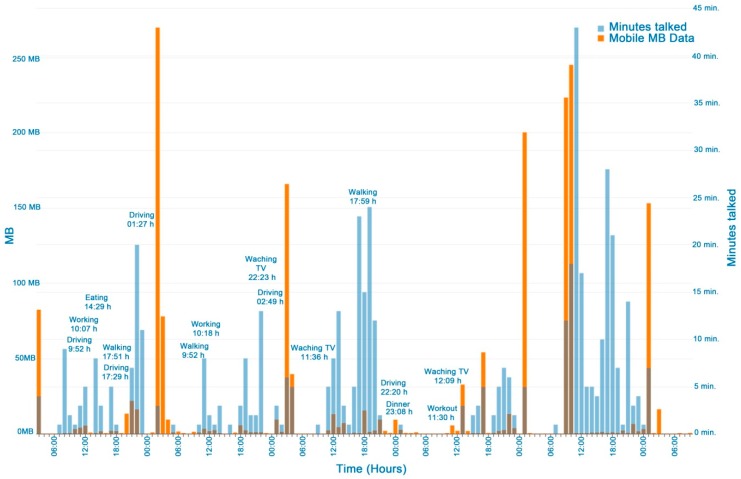
Data traffic (Mb) and time of talking or listening to sounds (minutes) by HDRS monitoring.

**Figure 8 ijerph-16-00461-f008:**
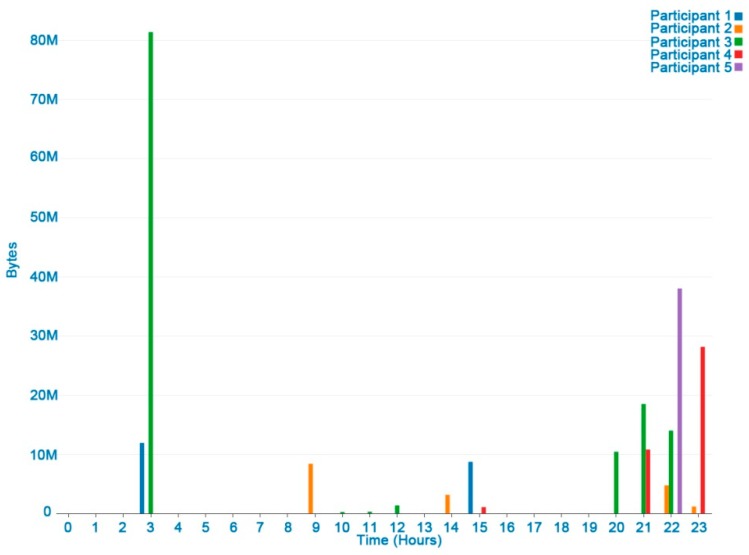
Traffic (Mb) in a day.

**Figure 9 ijerph-16-00461-f009:**
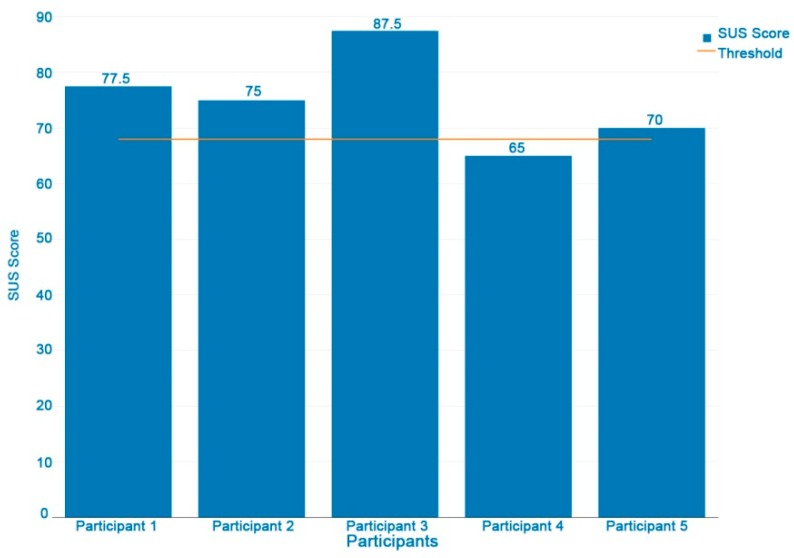
Graph of the results of the system usability scale (SUS) questionnaire. All the results exceed the threshold of 68 points (50th percentile) and represent an acceptable level of perceived usability of the system [27].

**Table 1 ijerph-16-00461-t001:** Preliminary relationships between the European Organization for the Research and Treatment of Cancer Quality of Life Questionnaire EORTC QLQ-C30 (Version 3.0) items and the Human Dynamics Reporting Service (HDRS) monitoring indicators.

EORTC QLQ-C30	Smartphone Data Sources
Constructed Scales(Period: During the Past Week)	Version 3.0Item Numbers	Amount of Movement (m/s^2^)	Distance from a Specific Place (Km)	Talk (0,1)	Data Taffic (MB)	The Number of Phone Calls (calls/min)
Global health status/QoL	
Global health status/QoL	29, 30					
Functional scales	
Physical functioning (strenuous activities, long/short walk; need to stay in bed; help with eating; dressing; washing or using the toilet)	1 to 5	●	●			
Role functioning (limited in work or other daily activities; limited in Hobbies/leisure time activities)	6, 7	●	●	●	●	●
Emotional functioning (feel tense/worry/irritable/depressed)	21 to 24					
Cognitive functioning (difficulty in concentrating on things/difficulty remembering things)	20, 25				●	
Social functioning (physical condition or medical treatment interfered with your family life/social activities)	26, 27			●	●	●
Symptom scales/items	
Fatigue (need to rest; felt weak; tired)	10, 12, 18					
Nausea and vomiting (nauseated; vomited)	14, 15					
Pain (felt pain/pain interfere with your daily activities)	9, 19					
Dyspnoea (short of breath)	8					
Insomnia (trouble sleeping)	11	●	●	●	●	●
Appetite loss (lacked appetite)	13					
Constipation (constipated)	16					
Diarrhoea (had diarrhoea)	17					
Financial difficulties (physical condition or medical treatment caused you financial difficulties)	28

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
