# Peer review of "Smartphone Sensors for Monitoring Cancer-Related Quality of Life: App Design, EORTC QLQ-C30 Mapping and Feasibility Study in Healthy Subjects"

_ijerph, 2019, doi:10.3390/ijerph16030461_

Round 1

Reviewer 1 Report

The ms presents a feasibility study of using mobile phone to capture the QoL of cancer patient. The intention was positive and has an important clinical implication. There are several issues that should be addressed prior to publication. 

Consideration of the novelty of the proposed app compared to the available tools need to be addressed. Currently, there are many apps developed to match unmet health care need. Is it possible to utilise or modify the existing one rather than develop a brand new one?  

Personal data protection needs to be addressed. In particular, when the sound and location  will be collected. 

how is it valid of using data traffic predict one's cognition? similar issue with social function? Correlation in such a small sample size may not be reliable. Also, there are other factors might be strongly correlated with QoL but not measured. The authors need to be aware of this. Furthermore, longer time of using Internet or cell phone have been correlated with greater level of depression. How this would affect the current findings?

It appears that only healthy subjects were included in this study. This is common practice for feasibility study. How this would apply to cancer patients? Would the same results be expected? 

when and how was QLQ-30 completed? 

Author Response

Dear Reviewer,

We sincerely appreciate your comments for the improvement of the article and our research, in the attached document you will find the answers to your comments.

Reviewer 2 Report

The authors have created a smartphone app with the aim to quantify and monitor cancer-related quality of life. They do this by continuously acquiring data streams from various sensors (accelerometer, GPS, microphone and phone calls) on the user's personal smartphone. They can then use the data to monitor the physical, psychological, social and environmental aspects of the user. The authors also provide a pilot test study with university volunteers. While not completely replacing the patient reported outcomes, the app has the potential to significantly reduce the resources and personnel required to continuously monitor cancer patients in clinical trials. Therefore I support the publication of this manuscript and have few comments/suggestions that the authors should address before publishing.

The authors describe in the introduction (page 1 and 2) the various aspects of smartphones that are now being harnessed in healthcare such as accelerometer, microphone,     speaker and camera. While there are many applications of these hardware components in the global health space, there is also a computational component that researchers are now tapping into on the smartphone for medical diagnostics. The continuously improving processors on the     smartphones are being used to provide computationally intensive     diagnostics

Rivenson, Yair,      Hatice Ceylan Koydemir, Hongda Wang, Zhensong Wei, Zhengshuang Ren, Harun      Günaydın, Yibo Zhang et al. "Deep learning enhanced mobile-phone      microscopy." ACS Photonics 5, no. 6 (2018): 2354-2364.

Priye, Aashish, Cameron S.      Ball, and Robert J. Meagher. "Colorimetric-luminance readout for      quantitative analysis of fluorescence signals with a smartphone CMOS      sensor." Analytical chemistry 90.21 (2018): 12385-12389.

The readers may benefit from a more complete introduction of smartphones with the inclusion of these applications as well.

Figure 7 caption: Distance traveled should be "m". "m/h", a unit of velocity is misleading here.

Author Response

(The authors gave the same response as above.)

Round 2

Reviewer 1 Report

The authors have addressed all comments. 

Author Response

No notes.